# Microstructures and Mechanical Properties of Al-2Fe-*x*Co Ternary Alloys with High Thermal Conductivity

**DOI:** 10.3390/ma13173728

**Published:** 2020-08-24

**Authors:** Gan Luo, Yujian Huang, Chengbo Li, Zhenghua Huang, Jun Du

**Affiliations:** 1School of Materials Science and Engineering, South China University of Technology, Guangzhou 510640, China; msluogan@mail.scut.edu.cn (G.L.); ms201720119373@mail.scut.edu.cn (Y.H.); mslcb@mail.scut.edu.cn (C.L.); 2Guangdong Provincial Key Laboratory of Metal Toughening Technology and Application, Guangdong Institute of Materials and Processing, Guangzhou 510650, China; 3Guangdong–Hong Kong Joint Research and Development Center on Advanced Manufacturing Technology for Light Alloys, Guangdong Institute of Materials and Processing, Guangzhou 510650, China

**Keywords:** Al-Fe alloy, Co modification, thermal conductivity, mechanical properties

## Abstract

The microstructures, mechanical properties, and thermal conductivity (TC) of Al-2Fe-*x*Co (*x* = 0~0.8) alloys in as-cast, homogeneous annealed, and cool rolled states are systematically studied. Results indicate that appropriate Co modification (*x* ≤ 0.5) simultaneously improves the thermal and mechanical properties of as-cast Al-2Fe alloys. The improvement of TC is attributed to ameliorating the morphology of primary Al_3_Fe phases from needles to short rods and fine particles, which decreases the scattering probability of free electrons during the electronic transmission. However, further increasing the Co content (*x* = 0.8) decreases the TC due to the formation of a coarse plate-like Al_2_FeCo phase. Besides, the thermal conductivity of annealed Al-2Fe-*x*Co alloys is higher than that of as-cast alloys because of the elimination of lattice defects and spheroidization of Al_3_Fe phases. After cool rolling with 80 % deformation, thermal conductivity of alloys slightly increases due to the breaking down of Al_2_FeCo phases. The rolled Al-2Fe-0.3Co alloy exhibits the highest thermal conductivity, which is about 225 W/(m·K), approximately 11 % higher than the as-cast Al-2Fe sample. The ultimate tensile strength (UTS) and elongation (EL) of as-cast Al-2Fe-0.5Co (UTS: 138 MPa; EL: 22.0 %) are increased by 35 % and 69 %, respectively, compared with those of unmodified alloy (UTS: 102 MPa; EL: 13.0 %).

## 1. Introduction

For the past decade, multiple researchers have tried to simultaneously improve the strength and electrical/thermal conductivity of aluminum alloys with the increasing demand of heat-dissipating equipment, such as heat radiators, 5G communication base stations, and so on [1,2,3,4]. However, it was reported that thermal conductivity and mechanical properties are contradictory factors in the alloys [5,6]. Aluminum alloys possessing higher thermal conductivity always exhibit poor mechanical properties and vice versa. The properties of aluminum alloys are controlled by certain factors, such as the alloying element [2,7], preparation process [8,9,10], heat treatment [10,11], etc. Moreover, the specific thermal resistivity (thermal resistivity increment of the alloy derived from unit mass addition) of the alloying elements for Al alloys varies. Specifically, alloying elements with low solid solubility and forming hard intermetallic phases are beneficial for enhancing the thermal conductivity and mechanical properties of alloys. This requirement can be satisfied by the Fe element; on the one hand, its solid solubility in α-Al is negligible [12], and on the other hand, the hard and brittle Fe-rich phases precipitate during the solidification process [13].

It is worth noting that Fe is inevitably adulterated in the production of die-casting aluminum alloys as anti-die-sticking elements, and its content is generally lower than 1.2 wt.% [13]. In addition, Al-Fe based alloys are widely used in the research and development of heat-resistant aluminum alloys [14]. To date, there are few studies that have investigated the relationship between thermal conductivity and Fe content in Al alloys. Chen et al. [7] showed that adding an appropriate amount of Fe element slightly improved the thermal conductivity of an Al-10Si alloy. Our research team pointed out that this improvement was attributed to the coupling effect between Fe and Si. However, the Si element significantly decreases the thermal conductivity of aluminum alloys [7,15]. For example, when the Si concentration reached 4 wt.%, the thermal conductivity of the Al-Si alloys dropped from 213.5 W/(m·K) to 165.1 W/(m·K) [7].

Therefore, the design and development of Si-free aluminum alloys are key to higher thermal conductivity. Generally, Fe usually leads to the formation of needle-shaped or flake-shaped Fe-rich intermetallic compounds. The hard and brittle Fe-rich phases are harmful for the mechanical properties because they could likely cause failure owing to decohesion, and the latent sites could initiate cracks [3,16]. Thus, controlling the morphologies and distributions of Al-Fe phases is the key to fabricate novel Al-Fe based alloys with high thermal conductivity and acceptable tensile properties. Over the years, some advanced processing techniques, such as electric-spark sintering [17], high-energy ball-milling [18], direct-current magnetic field treatment [19], rapid solidification [20], and liquid squeeze casting [21], were used to precipitate nanostructured Fe-rich phases and form fine α-Al grains. However, these techniques exhibit distinct difficulties for manufacturing large-scale and complex-shaped components.

To some extent, modification is a convenient method to control the intrinsic crystallization procedure of aluminum alloys, as well as ameliorate the morphology of second phases and refine the *α*-Al grains. It has been reported that Al-Ti-B, Al-Ti-C, or Al-Ti-C-B refiners can be used as effective modifiers to simultaneously refine Al_3_Fe phases and α-Al grains in hypereutectic Al-Fe alloys [22]. Moreover, it was demonstrated that the primary lath-shaped Al_3_Fe phase transformed into fine flowers and particles by adding 0.12 % Mg (mass ratio, same as below) to Al-5Fe melt [23]. Kaufman et al. [24] showed that Mn is an effective modified element, which completely converted the Fe phase from plate-like to Chinese script. However, the Mn element has a worse effect on the thermal conductivity than other elements because of its special configuration of an extra nuclear electron [5]. Thus, Mn is regarded as an impurity for high thermal conductivity aluminum alloys. It is well known that rare earth elements (RE) are commonly used modifiers. The addition of Ce-rich mischmetal changed the long needle-like Al_3_Fe phases into short rods and fine particles, which obviously improved the mechanical properties of hypoeutectic Al-Fe alloys [3]. Similarly, adding 0.3% RE was able to acquire the optimal microstructure and mechanical properties because of a reduction in the size of Fe-rich phases. When the RE content increased to 0.4%, the formation and aggregation of Al-Ce phases decreased the modification effect [25].

In terms of thermal conductivity, it has been estimated that the decreasing thermal conductivity caused by the alloying element in solid solution is approximately one order of magnitude larger than that of alloying elements forming intermetallic compounds [2]. Thus, choosing an element with low solid solubility as a modifier is an effective approach to simultaneously improve the thermal conductivity and mechanical properties of aluminum alloys. In this sense, the Co element with ignorable solid solubility in *α*-Al is one of the most effective elements to improve the morphology of Fe-rich intermetallics. Meng et al. [26] showed that adding 0.91% Co to Al-20Si-2Cu-1Ni-0.7Fe melt could effectively change the morphologies of Fe-containing compounds. Their morphology was mainly transformed from long acicular phases to Chinese script, granular, or rod-like Fe-containing phases resulting in the improvement of the tensile strength at room and elevated temperature. Some literature has been published on the mechanical properties of Co modification for Fe-containing Al-Si-based alloys [26,27]. No studies have yet shown a relationship between electrical/thermal conductivities, mechanical properties, and the microstructure of near-eutectic Al-2Fe alloys with different Co contents. In our study, Al-Fe based alloys are the potential material to develop high thermal conductivity Al alloys with acceptable mechanical properties. A near-eutectic Al-2Fe alloy was set as the research object due to its excellent castability, and the effects of Co modification on the microstructures, mechanical properties, and electrical/thermal conductivity of the Al-2Fe alloy were systematically investigated. The microstructures and corresponding performances for their as-cast, homogenous annealed, and rolled states were observed and tested. Finally, the Co modification behaviors of the Al-2Fe alloys and the modification mechanism were further discussed. This study will help to provide the theoretical basis to develop novel Al-Fe-based wrought aluminum alloys with high thermal conductivity and acceptable mechanical properties.

## 2. Materials and Methods

### 2.1. Preparation of Samples

The Al-2Fe-*x*Co (*x* = 0~0.8) alloy ingots were melted by commercial pure aluminum ingot (99.8% Al), Al-20%Fe, and Al-10%Co master alloys. The pure aluminum ingots were melted in the graphite clay crucible by an electric resistance furnace at 1023 K. The Al-20%Fe master alloy was then added to form the Al-2%Fe alloy. The melt was stirred with a MgO ceramic rod for approximately 1 min to ensure the uniformity of the melt. Definite amounts of the Al-10%Co modifier were added to form five groups of samples containing 0%, 0.1%, 0.3%, 0.5%, and 0.8%, respectively. When the temperature of the melt decreased to 993 K, the melts were poured into a steel mold preheated to 473 K. The ingots with the dimension of 100 mm × 45 mm × 15 mm were cooled to room temperature in the mold.

The as-cast samples were longitudinally cut into three parts at 40 mm and 80 mm away from the right. The middle and right samples were homogenized at 773 K for 24 h, and the annealed right one was rolled from a thickness of 15 mm to 3 mm with 12 passes at ambient temperature. The deformation degree of each pass was approximately 1 mm. The rolling process of the Al-2Fe-*x*Co alloys is schematically presented in Figure 1.

The total rolling deformation was 80%. The as-cast, annealed, and rolled plates were machined into tensile samples. All tensile tests were conducted at ambient temperature.

### 2.2. Measurements

Metallographic specimens were prepared by grinding, polishing, and etching in 0.5% (volume fraction) aqueous hydrofluoric acid (HF) solution. The microstructures of the as-cast, annealed, and rolled samples were observed by an optical microscope (OM, Leica DMI 3000, Leica, Germany) and scanning electron microscope (SEM, Zeiss Gemini 300, Carl Zeiss, Germany). Energy dispersive spectrum (EDS, Oxford X-MaxN, Oxford, UK) was used to determine the second phase composition. The constituent phases of the samples were identified by X-ray diffraction (XRD, Bruker D8 Advance, Bruker, Germany) with Cu-K_α_ radiation. A material test machine (AG-X100kN, Shimadzu, Japan) was used to examine the tensile properties at the loading speed of 1.0 mm/min. The hardness (HB) was tested using a hardness tester (HB-3000, Shanghai, China). Three samples for each group were used for obtaining the mechanical properties of each state alloy.

All conductivity performance tests were carried out at ambient temperature. The thermal diffusivities of the cylindrical samples with the size of Φ12.7 × 3 mm^2^ in round disks were measured by the flash method (Netzsch LFA457, Netzsch, Germany). The densities of samples were determined by the Archimedes method (DH-300, Shenzhen, China). The specific heat capacities of the alloys were calculated using the Neumann–Kopp rule [28,29]. Thus, the thermal conductivity (*λ*) of the sample was calculated by following Equation (1):(1)$$\lambda =\alpha \cdot \rho \cdot C_{p}$$
where *α* is the thermal diffusivity (cm^2^/s), *ρ* is the density (g/cm^3^), and *C_p_* is the specific heat capacity (J/(g·K)). The error in the thermal conductivity measurement was less than ±5%. Each test was repeated three times for each sample, and the average value was taken to ensure the reliability of the experiment. Moreover, to determine the contribution of free electrons in the heat transferring processing, electrical conductivity measurement was conducted on the samples by the vortex method (FD-101, Xiaman, China).

## 3. Results and Discussion

### 3.1. XRD Results

The X-ray diffraction testing was conducted for the as-cast Al-2Fe and Al-2Fe-0.8Co alloys. The analysis results are shown in Figure 2.

The as-cast Al-2Fe eutectic alloy consists of cubic α-Al (PDF# 00-001-1180 [30]) and monoclinic Al_3_Fe (Al_13_Fe_4_, PDF# 00-050-0797 [31]) phases. The Co modification introduces cubic Al_2_FeCo phases (PDF# 03-065-4920 [32]).

The Co modification changes the relative intensities of the diffraction peaks. The intensity of the close-packed (111)_Al_, (220)_Al_ and (222)_Al_ plane increases. These results suggest that Co modification disturbs the normal crystallization process and refines the grain of α-Al. The growth dependent on the (111)_Al_, (220)_Al_ and (222)_Al_ plane is promoted, which is beneficial for the uniformity of the microstructure. This result is in agreement with the optical microstructure shown in Figure 3.

### 3.2. Microstructure Characterization

Figure 3 presents the optical micrographs (OM) of the as-cast Al-2Fe-*x*Co alloys. The size of the α-Al grains first decreases and then increases with the increase in the Co content. From Figure 3a, it can be observed that the primary Fe-containing intermetallic compounds are presented by long needles in the range of 20~40 μm in length, while the eutectic Al-Fe phases are presented along the *α*-Al grains for unmodified samples. The addition of Co obviously changes the microstructure of the alloy, as shown in Figure 3b–e. First, the length of the primary Fe-containing phases gradually decreases with increasing Co content. When the amount of added Co reaches 0.3 wt.%, the optimal microstructure is obtained, where the primary Fe-containing intermetallic compounds are prominently transformed from long needles to fine particles. However, when the Co content further increases to 0.8 %, the coarse plate-like phases are presented in the α-Al matrix. From the EDS results shown in Figure 4c, the intermetallic compound is the Al-Fe-Co ternary phase.

The atom ratio between the Fe and Co elements is close to 1:1. Combined with the results of XRD analysis as shown in Figure 2b, the Al-Fe-Co ternary phase is thought to be Al_2_FeCo. As for the eutectic Al-Fe phase, it is refined and transformed into a discontinuous network owing to divorced eutectics by Co modification.

Based on the solidification principle [33,34,35], the grain refinement depends on the supercooling effect of the melt during the crystallization process. The critical condition of the supercooling relies on the ratio between the concentration gradient at the liquid–solid interface and the growth rate of the grains. This ratio is connected to the slope of corresponding composition, the solute distribution coefficient, and the composition of the alloy. Combined with the Al-Co binary phase diagram [12], the liquidus decreases with increasing Co content from 0% to 1.8%. In this study, the slope of the corresponding composition and the solute distribution coefficient are regarded as constants. Therefore, the degree of supercooling mainly depends on the composition of the alloy. The above deduction could explain the grain refinement behavior observed in Figure 3. The constitutional supercooling extent of the melt becomes more prominent when the Co content is in the range from 0 to 0.3%. Within this range, a higher amount of secondary phases will further inhibit the growth of *α*-Al grains.

However, when the Co content is higher than 0.5%, the grains become coarse. This is due to the formation of the Al_2_FeCo phase, which weakens the extent of constitutional supercooling. Figure 4 shows the SEM-SE micrographs of as-cast Al-Fe-*x*Co alloys and corresponding EDS results. The primary Fe-rich phases are presented as long needles (Point A in Figure 4a), while the eutectic Fe-rich phases exist as fine particles (Point B in Figure 4a). As shown in Figure 4b Point C, the Co element is not detected in the matrix. Adding 0.3% Co transforms the primary Fe-containing intermetallic compounds into short rods (Point D in Figure 4b). Co modification has little effect on the eutectic Al_3_Fe phases, which are still presented as particles in the Al matrix (Point E in Figure 4b). However, when the Co content increases to 0.8%, coarse plate-like Al_2_FeCo phases are generated with a size of about 25~40 μm in length and 6~10 μm in width (Point F and G in Figure 4c).

As shown in Figure 5, after the homogeneous annealing, the primary and eutectic Al-Fe phases decompose into short flakes or particles, while the size and morphology of Al_2_FeCo phases are hardly changed (Figure 5e).

The microstructures of rolled Al-2Fe-*x*Co alloys are shown in Figure 6.

The intermetallic compounds uniformly distribute in the *α*-Al matrix. Under the rolling deformation of 80%, the long needle-like Fe-containing phases break up and transform into fine particles when the Co content is in the range from 0 to 0.5 wt.%. Moreover, when the Co content reaches 0.8 wt.%, the sizes and morphologies of Al_2_FeCo prominently decrease and transform from plates to fine particles and short rods during the rolling deformation process.

### 3.3. Conductivity Performance of Al-2Fe-xCo Ternary Alloys

The conductivity performance of the Al-2Fe-*x*Co alloys in different states is shown in Figure 7.

From Figure 7a, the thermal diffusivity of as-cast specimens first increases from 0.837 cm^2^/s for the Al-2Fe alloy to 0.856 cm^2^/s for Al-2Fe-0.3Co alloy by improving the amplitude of 2.3%, and then gradually decreases with increasing Co content. The annealed and rolled alloys exhibit higher thermal diffusivities. Their variations are similar to those of the as-cast samples. In the case of rolled alloy samples, the highest TC achieved is 0.925 cm^2^/s for the Al-2Fe-0.3Co alloy, about 10.5% higher than the as-cast Al-2Fe alloy. As shown in Figure 7b, the density of these alloys increases with the increase in the Co content. The density of the samples in the rolled state is highest, followed by the as-cast ones, and the lowest in the annealed ones. In addition, the specific heat capacity, calculated by the Neumann–Kopp rule [28,29], decreases linearly with increasing Co content.

The tendency of TC calculated by Equation (1) (as shown in Figure 7c) for these specimens in different states is similar to that of thermal diffusivity. In the as-cast state, the optimal TC, about 208 W/(m·K), appears in the range of the Co content from 0.3% to 0.5% with an increase of 2.2% relative to the Al-2Fe alloy (about 203 W/(m·K)). The homogenization and rolling deformation are beneficial to the conductivity performance. The maximum thermal conductivities in annealed and rolled states reach 225 W/(m·K) at the same time, which is approximately 10.8% higher than the as-cast Al-2Fe sample. However, further increasing the Co content will decrease the TC of Al-2Fe alloys irrespective of the alloy state. To determine the contribution of free electrons in the heat transfer process, electrical conductivities are measured as shown in Figure 7d. It can be observed that the electrical conductivity and thermal conductivity are positively correlated, and their mathematical relationship will be deeply discussed.

According to Figure 7, it is worth noting that the conductivity performance of different states for Al-2Fe-*x*Co alloys are in the following sequence: as-cast < annealing < rolling. It is known that many factors affect the thermal conductivity of alloys, such as alloy composition [2,7,36,37], heat treatment [10,11,38,39], melt treatment [40,41], plastic deformation [1,42], and so on. In our study, compared with the as-cast samples, the thermal conductivities of alloys significantly increase after annealing treatment due to the evolution of the morphology and elimination of lattice defects. The long needle-like Fe-containing phases transform into fine particles and short flakes, reducing the scattering influence of free electrons during heat transfer. It has been estimated that homogenization could effectively reduce the vacancy concentration in the Al matrix [43]. Vacancy, a kind of lattice defect, is taken as a strong scatter source [44]. Thus, decreasing the vacancy concentration of the matrix is able to improve the thermal conductivity of alloys. In conclusion, the enhancement of thermal conductivity for annealed Al-2Fe alloys with different Co contents is attributed to the decrease of vacancy concentration and the morphological improvement of Fe-containing compounds. 

Compared with the annealed alloys, rolling deformation could slightly increase the thermal conductivity of Al-2Fe-*x*Co alloys. The extent of enhancement was relatively low compared with annealed alloys. It has been reported that plastic deformation for metal can reduce the number of macro defects, such as shrinkage cavity and porosity [45]. Therefore, rolling deformation increased the density of alloys, resulting in the slight increase in conductivity performance. For metal and alloys, the thermal conductivity is in proportion to the electrical conductivity according to the Wiedemann–Franz law, Equation (2) [46]:(2)λ=LTσ
where *λ* denotes thermal conductivity (W/(m·K)), σ is electrical conductivity (MS/m, M = 10^6^), *T* is temperature in Kelvin (K), and *L* is the Lorentz number (*L* = 2.44 × 10^−8^ V^−2^K^−2^). As shown in Figure 8, the blue dotted line is the linear relationship between thermal conductivity and electrical conductivity according to the Wiedemann–Franz law.

Linear fitting (black dotted line) is conducted to match this relationship for Al-2Fe-*x*Co ternary alloys in the different states as shown by the black dotted line.

The best fitting effect is obtained when the intercept is 6.96. In other words, the Lorentz number (*L*) is equal to 2.34 × 10^−8^ V^−2^K^−2^ for the Al-2Fe-*x*Co alloys. 

Based on these data, the Wiedemann–Franz law estimations are notably higher than the measured thermal conductivities for the Al-2Fe-*x*Co alloys. The discrepancy can be considered as a constant, approximately 10 W/(m·K) in this study. Similar phenomena for Al-Si based alloys are reported by Chen [7] and Hatch [47], and relative adjustment terms are proposed. A possible explanation for the discrepancy could be the precipitates of Al_3_Fe and Al_2_FeCo with poor thermal conductivity. These intermetallic compounds reduce the free paths for electron migration and thus the contribution of electron conduction. Moreover, it is evident that the correlation between thermal conductivity and electrical conductivity is not linked with heat treatment and plastic deformation.

The classical thermal conductivity theory demonstrates that the heat conduction of metals mainly depends on the electron conduction. Based on the Drude theory [48,49], the electrical conductivity is proportional to the relaxation time (*τ*) of free electrons during electronic transmission, Equation (3):(3)$$k_{e}\;=\;\frac{1}{3}n\nu^{2}\tau c_{\nu }\;=\;\frac{1}{3}n\nu lc_{\nu }$$
where *n* is the number of effective free electrons, *ν* is the average speed of free electrons, *l* is the mean free path of electron movement, and *c*_v_ is the specific heat of the metal. Generally, *ν* and *τ* can be regarded as constant. Therefore, the heat conduction is determined by the number of effective free electrons (*n*) and the mean free path of conduction electrons (*l*).

As for the unmodified alloy, the primary Al_3_Fe phases are presented as long needles, which form a series relationship with the Al matrix. In other words, the free electron must pass through the Al_3_Fe phases with higher electrical resistance. After Co modification, the morphologies of primary Al_3_Fe phases transform from long needles to short rods and fine particles. The rod-shaped or particle-shaped Al_3_Fe phases maintain a parallel relationship with the Al matrix. That means the increasing connectivity of the Al matrix augments the free path of electron movement (*l*) and the number of effective free electrons (*n*). Combined with the Drude theory [48,49], the decreasing scattering probability of free electrons during the heat transfer process results in the improvement of electrical/thermal conductivity. Similarly, the coarse plate-shaped Al_2_FeCo phases for the as-cast Al-2Fe-0.8Co alloy increase the scattering probability of free electrons and decrease the conductivity performance. 

### 3.4. Mechanical Properties of Al-2Fe-xCo Ternary Alloys

The ultimate tensile strength (UTS), elongation (EL), and hardness (HB) of the Al-2Fe-*x*Co alloys in various states are shown in Figure 9.

The UTS, EL, and HB of as-cast Al-2Fe alloys are 102 MPa, 13.0%, and 36.5 HBW, respectively. With increasing Co content, the mechanical properties increase obviously at first, and then decrease. When the Co content is 0.5 %, the comprehensive mechanical properties reach the optimum, with UTS, EL, and HB of 138 MPa, 22.0%, and 40.5 HBW, respectively. The improvement of mechanical properties is due to the grain refinement and second phase strengthening [3,16].

After homogenization, these alloys possess higher elongation. The maximum value of EL can reach 25.0% while the Co content is 0.5%. However, the UTS and HB of annealed alloys are lower than those of as-cast alloys. In addition, the UTS and HB significantly increase through rolling deformation. The optimal UTS and HB are 192 MPa and 50 HBW for cool-rolled Al-2Fe-0.5Co alloys, which are 88.2% and 37.0% higher, respectively, than the as-cast Al-2Fe alloy. However, the EL of rolled samples inevitably decreases. 

The mechanical properties of eutectic Al-Fe alloys mainly depend on the grain of α-Al and the size, morphology, and distribution of secondary phases. The hard primary Al_3_Fe phases exist as long needles (Figure 3a) in the as-cast Co-free alloy, which significantly reduces the mechanical properties of the Al-2Fe alloy. Its mechanical properties are hardly improved by heat treatment (Figure 5 and Figure 3a). 

The positive influence of Co modification on the mechanical properties of as-cast Al-2Fe alloys could be mainly attributed to the grain refinement and second phase strengthening. The proper addition of Co simultaneously refines the grains of *α*-Al and primary Al_3_Fe phases. The uniform microstructures are formed by Co modification. Homogenization generates more fine particles primary Al_3_Fe phases, which are uniformly distributed in the aluminum matrix (see details in Figure 5b,c). The rolling process decreases the number of macro defects and increases the density of dislocation, which changes the fracture mechanism.

According to the mentioned explanation, the optimal microstructure for as-cast Al-2Fe alloys with various Co amounts is obtained when the Co content is 0.3%. However, the Al-2Fe-0.5Co alloy exhibits the best mechanical properties. Combined with the microstructure given in Figure 3d, some fine rod-like Al_2_FeCo phases are generated in the Al matrix. Short rod-like Al_2_FeCo phases could be used as a strengthening phase, which further increases the mechanical properties. Conversely, when the Co content reaches 0.8%, the grain coarsening and plate-like Al_2_FeCo phases decrease the mechanical properties. The formation of Al_2_FeCo phases significantly weakens the extent of constitutional supercooling, which coarsens the α-Al grains. Moreover, the sharp plate-like Al_2_FeCo phases easily become the origin of cracks, which decreases the mechanical properties. 

### 3.5. The Correlation between Thermal Conductivity and Mechanical Properties

According to the statistical data [50] in Figure 10, the strength-conductivity balance of the Al-2Fe-0.3Co alloy is in comparison with several present commercial wrought aluminum alloys.

Obviously, thermal conductivity (TC) and ultimate tensile strength (UTS) show a negative correlation for the commercial wrought aluminum alloys. It can be observed that 1xxx series aluminum alloys possess higher TC, but their UTS is relatively poor due to the low content of alloying elements [41]. 2xxx and 7xxx series aluminum alloys with excellent mechanical properties are widely used in the aerospace field [51]. However, their thermal conductivities are difficult to meet the demand of heat-dissipating equipment. It is known that 6xxx series aluminum alloys are widely used in the electronic communication field [52,53]. Their thermal conductivity and mechanical properties could be further improved. 

In this study, we successfully developed a novel Al-Fe based alloy with high thermal conductivity (about 225 W/(m·K)) and acceptable ultimate tensile strength (near 200 MPa). It is worth mentioning that the thermal conductivity of this novel alloy is close to that of 1xxx series aluminum alloys, and the mechanical properties are higher than those of 6xxx series aluminum alloys. This comparison shows that the tensile strength is not particularly high. Despite this, the alloy exhibits a better combination of medium strength and high thermal conductivity than 6xxx series aluminum alloys. The combination of high-conductivity and medium strength exhibited by the rolled Al-2Fe-0.3Co alloy is a promising result for its use in industrial applications, such as heat radiators and 5G communication base stations. Furthermore, the thermal conductivity exhibited by the rolled Al-2Fe-0.3Co alloy increases by 3% and 25 % when compared to the respective 6063 (218 W/(m∙K)) and 6061 (180 W/(m·K)) commercial wrought aluminum alloys.

## 4. Conclusions

According to the results of this work, it is found that the mechanical properties and conductivity performance of the Al-2Fe alloy are simultaneously improved by Co modification. This finding is favorable to develop structural aluminum alloys with the demands of low processing cost and sufficient thermal conductivity. Several significant conclusions can be drawn as follows:(1)The addition amount of Co in the range from 0 to 0.3% can transform the morphology of primary Al_3_Fe phases from long needles to fine particles. The thermal conductivity of the Al-2Fe matrix would slightly increase from 203 W/(m·K) to 208 W/(m·K).(2)Because of the elimination of lattice defects and spheroidization of Al_3_Fe phases, the thermal conductivity of annealed Al-2Fe-*x*Co alloys is higher than that of as-cast alloys. After cool rolling with 80% deformation, the thermal conductivity of alloys slightly increases due to the breaking down of Al_2_FeCo phases.(3)Linear fitting was conducted to match this relationship for Al-2Fe-*x*Co ternary alloys in different states. The best fitting effect was obtained when the intercept was 6.96. In other words, the Lorentz number (L) is equal to 2.34 × 10^−8^ V^−2^K^−2^ for the Al-2Fe-*x*Co alloys.(4)The UTS and EL of the Al-2Fe-0.5Co alloy were close to 140 MPa and 22.0%, respectively, i.e., about 35% and 69% higher than those of the matrix Al-2Fe alloy. The improvement of mechanical properties was attributed to the refinement of α-Al grains and second phase strengthening.

## Figures and Tables

**Figure 1 materials-13-03728-f001:**
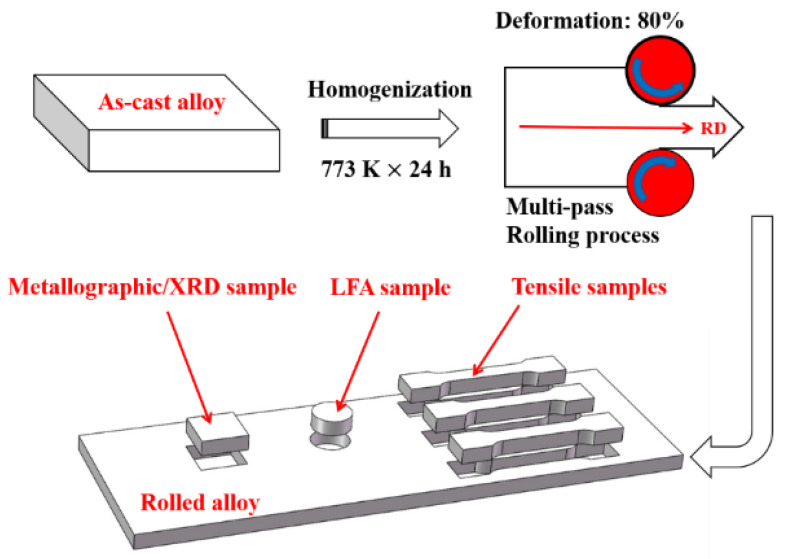
A schematic diagram of the rolling process and position of the metallographic sample, LFA (Laser flash) sample, and tensile samples obtained from the rolled Al-2Fe-*x*Co alloys.

**Figure 2 materials-13-03728-f002:**
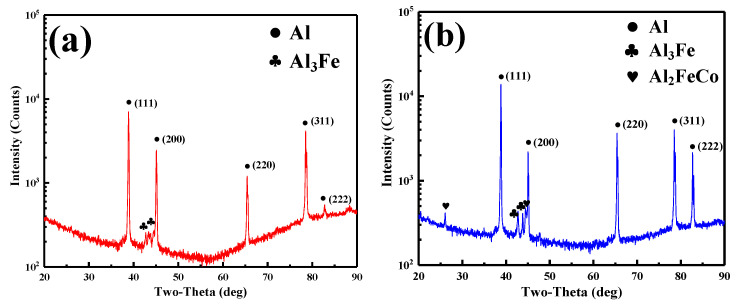
XRD patterns of the alloys subject to different conditions: (**a**) as-cast Al-2Fe; (**b**) as-cast Al-2Fe-0.8Co.

**Figure 3 materials-13-03728-f003:**
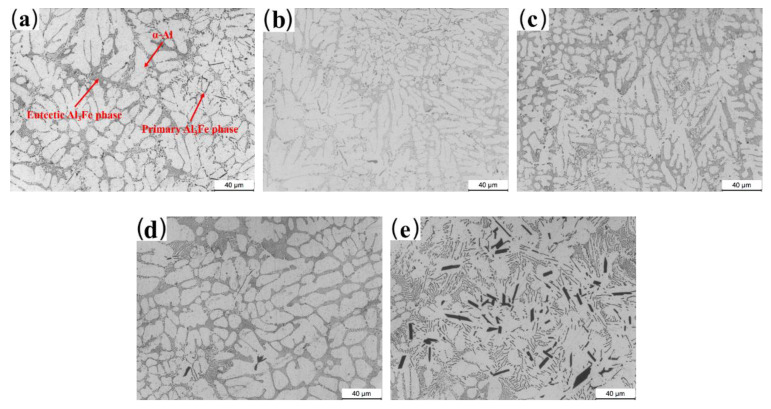
Microstructure of the Al-2Fe alloys modified with different contents of Co in the as-cast state: (**a**) Co-free; (**b**) 0.1%; (**c**) 0.3%; (**d**) 0.5%; (**e**) 0.8%.

**Figure 4 materials-13-03728-f004:**
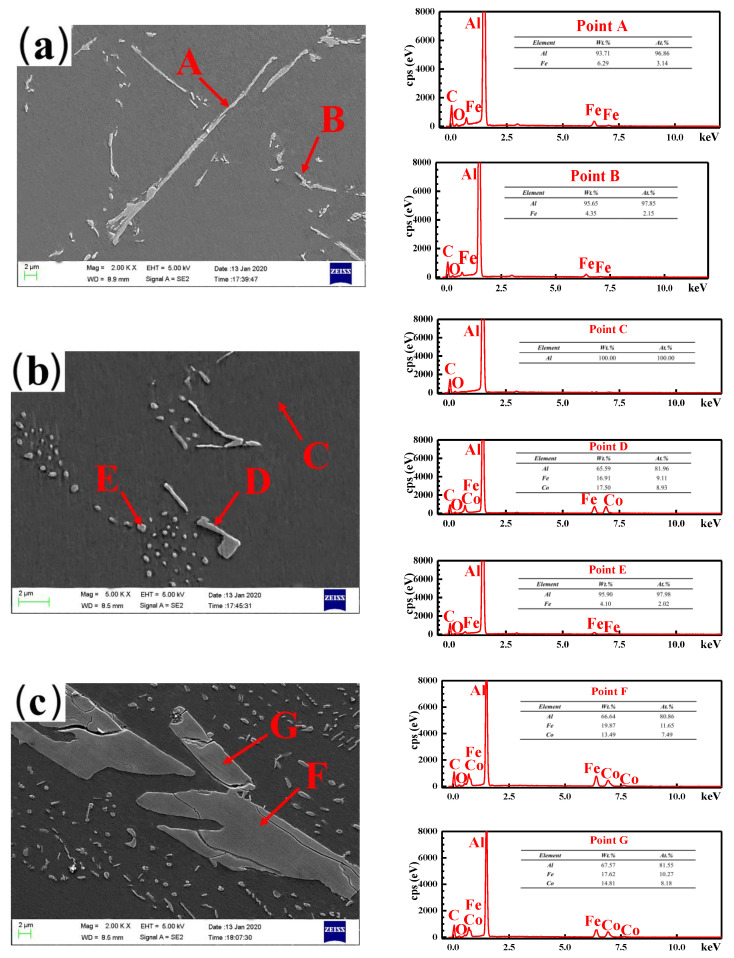
SEM-SE images of as-cast (**a**) Al–2Fe, (**b**) Al–2Fe-0.3Co, (**c**) Al-2Fe-0.8Co alloys, and corresponding EDS results.

**Figure 5 materials-13-03728-f005:**
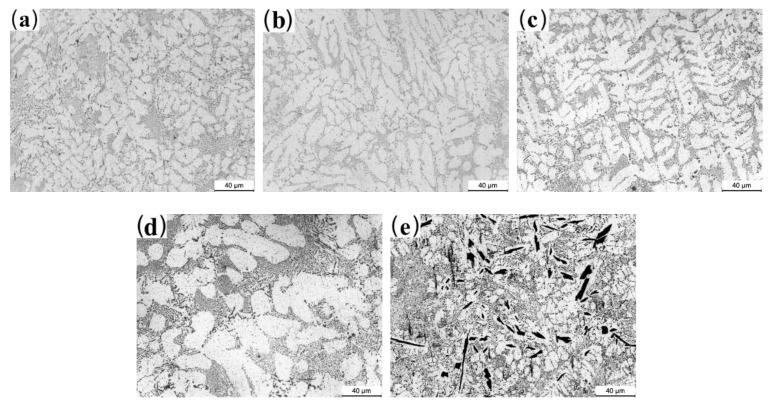
Microstructure of the annealed Al-2Fe-*x*Co alloys: (**a**) Co-free; (**b**) 0.1%; (**c**) 0.3%; (**d**) 0.5%; (**e**) 0.8%.

**Figure 6 materials-13-03728-f006:**
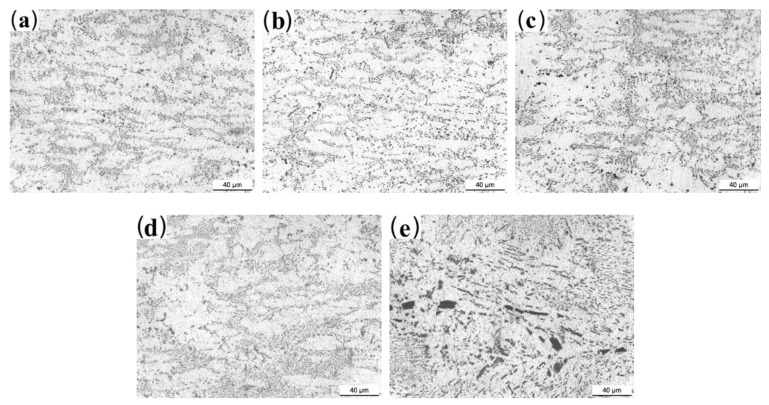
Microstructure of the rolled Al-2Fe-*x*Co alloys: (**a**) *x* = 0; (**b**) *x* = 0.1; (**c**) *x* = 0.3; (**d**) *x* = 0.5; (**e**) *x* = 0.8.

**Figure 7 materials-13-03728-f007:**
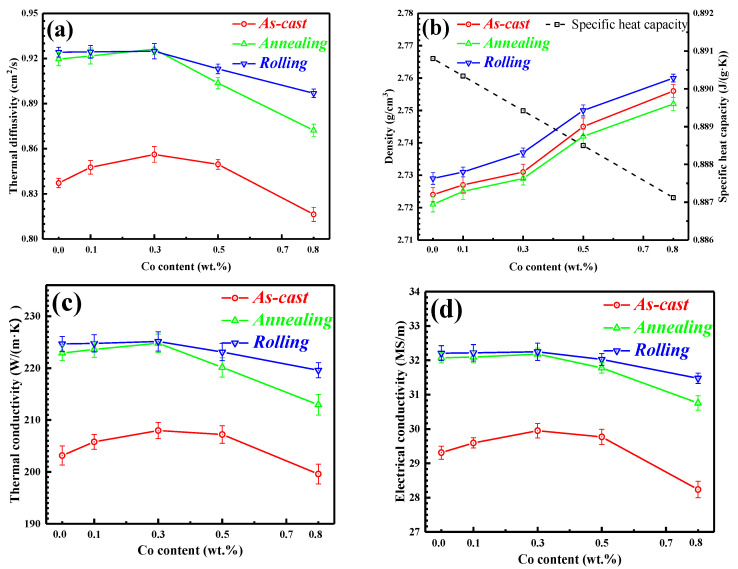
The various properties of the Co-modified Al-2%Fe alloys in different states; (**a**) Thermal diffusivity; (**b**) Density and specific heat capacity; (**c**) Thermal conductivity; (**d**) Electrical conductivity.

**Figure 8 materials-13-03728-f008:**
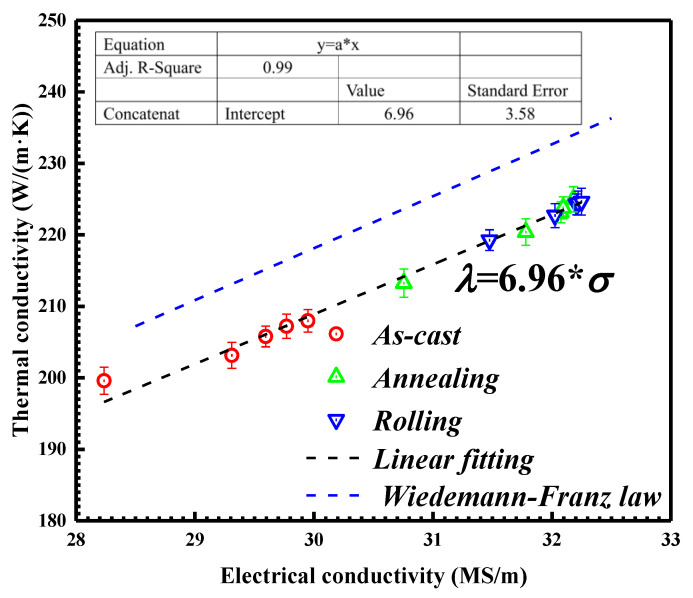
The linear fitting between electrical conductivity and thermal conductivity of Co-modified Al-2%Fe alloys in different states.

**Figure 9 materials-13-03728-f009:**
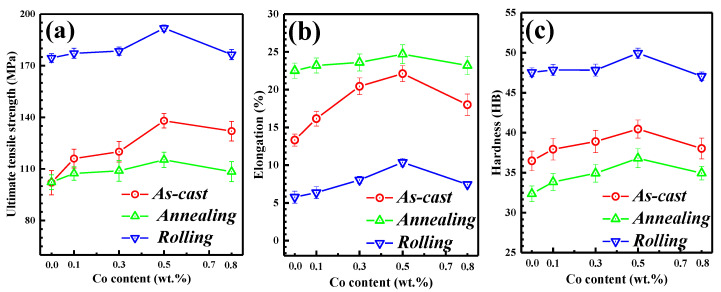
The mechanical properties of Al-2Fe with different Co contents in different states: (**a**) Ultimate tensile strength, (**b**) Elongation, (**c**) Hardness.

**Figure 10 materials-13-03728-f010:**
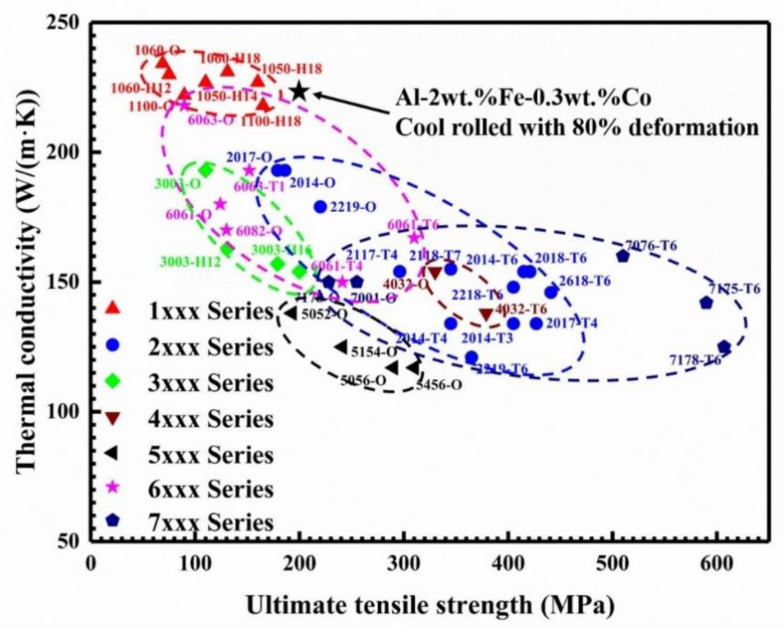
The correlation between the thermal conductivity and ultimate tensile strength of commercial wrought aluminum alloys.

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
