# Peer review of "Microstructures and Mechanical Properties of Al-2Fe-xCo Ternary Alloys with High Thermal Conductivity"

_materials, 2020, doi:10.3390/ma13173728_

Round 1

Reviewer 1 Report

General comments:

- In general, the research and results are relevant and interesting.

- It is not well written, it contains numerous grammar errors.  General revision of the English is required.

- The discussion about the effect of Co on the mechanical properties of the alloy should be improved.

Specific comments:

  1. Introduction

This section presents multiple grammar errors and requires an extensive revision of the English language. Some examples are:

Page 1 - Line 32: “Since now, multiple… ” Since now is not the correct way to start this sentence.

Page 1 - Line 39: Rewrite the following sentence, the grammar is not right: “Specially, the alloying element with low solid solubility and forming a hard second phase in Al matrix is the ideal candidate to prepare high thermal conductivity alloys ”.

Page 1 - Line 41 and Page 2 – Line 42: Two consecutive sentences starting with “On the other hand,….” it is not acceptable in English.

Page 2 – Line 73: Rewrite the following sentence, the grammar is not right: “However, the decrease thermal conductivity caused by Mn was quite larger than other elements because of its special configuration of extra-nuclear electron.”

  1. Materials and methods

The section is not well written with multiple grammar errors and with some points that should be clarified:

-        Which  is  the annealing treatment? 24 hours at 773 K? If so, it should be clearly stated that this is  the annealing treatment that the results refer to.

-        When were the samples taken after the annealing treatment and how? In this section it is only mentioned when they were taken, the methodology for the initial samples and the samples after rolling, but it is not said where and when the annealed samples presented in Figure 5 were taken.

-        It is stated that 5 different Co levels (0%, 0.1%, 0.3%, 0.5% and 0.8%) have been produced and studied, but no results are presented for the 0.1% Co level. Why? Either remove this level from the “Materials and Methods” section, or present the results for this composition level.

Page 3 - Line 106: Rewrite the following sentence, it is not clear: “With the weight of about 300 g aluminium ingots were melted at 1023 K in the graphite clay crucible by electric resistance furnace”.

Page 3 - Line 109: Rewrite the following sentence, the grammar is not right: “Subsequently, various content of the Al-10%Co modifier were added to prepare five sample groups with the Co content of 0%, 0.1%, 0.3%, 0.5% and 0.8%, respectively”.

Page 3 – Line 139: The units of the formula do not fit. Please, adjust the units (and formula if required) in order that the different variables fit together.

Figure 2. The figure a is missing in the article! (at least in the version I have )

  1. Results and Discussion

Page 7 – Line 181: .b ¿?

Page 10 – Line 249: Rewrite the following sentence, it is confusing and not well written: “In our study, after homogenization, the thermal conductivity significantly increases due to the morphological evolution and eliminate of lattice defects”.

Page 13 – Line 308: “With increasing the Co content, the mechanical properties increase obviously at first, and then decrease”. Why it is so obvious? I do not see how it is so obvious At least include a reference that report that change of mechanical properties with Co content in Al-2Fe alloys.

Page 14 – Line 336: “The correlation between conductivity and mechanical property”, should be “….mechanical properties”.

Author Response

Response to Reviewer 1 Comments

Point 1: Page 1 - Line 32: “Since now, multiple… ” Since now is not the correct way to start this sentence.

Response 1: “Since now” has been revised to “For the past decade” (see Page1-Line 32 in the revised manuscript)

Point 2: Page 1 - Line 39: Rewrite the following sentence, the grammar is not right: “Specially, the alloying element with low solid solubility and forming a hard second phase in Al matrix is the ideal candidate to prepare high thermal conductivity alloys”.

Response 2: This sentence has been verified to “Specially, the alloying elements having low solid solubility in Al matrix and forming hard intermetallic phases are beneficial for enhancing the thermal conductivity and mechanical properties of alloys.” (see Page1-Line 40 in the revised manuscript)

Point 3: Page 1 - Line 41 and Page 2 – Line 42: Two consecutive sentences starting with “On the other hand,….” it is not acceptable in English.

Response 3: This sentence does be not acceptable in English. It has been verified to “Among them, this requirement can be satisfied by Fe element, on the one hand, its solid solubility in α-Al is ignorable [12], on the other hand the hard and brittle Fe-rich phases precipitate during the solidification process [13].” (see Page1-Line 42 in the revised manuscript)

Point 4: Page 2 – Line 73: Rewrite the following sentence, the grammar is not right: “However, the decrease thermal conductivity caused by Mn was quite larger than other elements because of its special configuration of extra-nuclear electron.”

Response 4: The sentence has been revised to “However, Mn element has a worse effect on the thermal conductivity than other elements because of its special configuration of extra-nuclear electron [5].” (see Page 2-Line 73 in the revised manuscript)

Point 5: Which is the annealing treatment? 24 hours at 773 K? If so, it should be clearly stated that this is the annealing treatment that the results refer to.

Response 5: Thank you for pointing out the unclear statement. In this study, the temperature and hold time of annealing treatment is 773 K and 24 h, respectively. Thus, the corresponding sentences have been verified to “The as-cast samples were longitudinally cut into three parts at 40 mm and 80 mm away from the right. The middle and right ones were homogenization treated at 773K for 24 h, and the annealed right one was rolled from a thickness of 15 mm to 3 mm with 12 passes at ambient temperature.” (see Page 3-Line 114)

Point 6: When were the samples taken after the annealing treatment and how? In this section it is only mentioned when they were taken, the methodology for the initial samples and the samples after rolling, but it is not said where and when the annealed samples presented in Figure 5 were taken.

Response 6: Thank you for this valuable comment. The annealed samples were taken in the middle ingot. After the middle ingot treated by homogenization, the corresponding testing samples were machined in the annealed ingot. To clearly state the annealing treatment, the related sentences have been verified to “The as-cast samples were longitudinally cut into three parts at 40 mm and 80 mm away from the right. The middle and right ones were homogenization treated at 773K for 24 h, and the annealed right one was rolled from a thickness of 15 mm to 3 mm with 12 passes at ambient temperature.” (see Page 3-Line 114)

Point 7: It is stated that 5 different Co levels (0%, 0.1%, 0.3%, 0.5% and 0.8%) have been produced and studied, but no results are presented for the 0.1% Co level. Why? Either remove this level from the “Materials and Methods” section, or present the results for this composition level.

Response 7: Thank you for the good advice. The results of Al-2Fe-0.1Co and corresponding sentences and figures were presented in the revised manuscript:

       “Fig. 3b-d” has been revised to “Fig. 3b-e” (see Page 6-Line 170)

       “Fig. 5d” has been revised to “Fig. 5e” (see Page 7-Line 207)

       “Fig. 5b” has been revised to “Fig. 5b and c” (see Page 12-Line 328)

“Fig. 5c” has been revised to “Fig. 5d” (see Page 12-Line 333)

The optical microstructures of as-cast, annealed and rolled Al-2Fe-0.1Co have been added into Figure 3b, 5b and 6b.

Point 8: Page 3 - Line 106: Rewrite the following sentence, it is not clear: “With the weight of about 300 g aluminium ingots were melted at 1023 K in the graphite clay crucible by electric resistance furnace”.

Response 8: This sentence has been verified to “The commercial pure aluminum ingots were melted in the graphite clay crucible by electric resistance furnace at 1023K.” (see Page 3-Line 107)

Point 9: Page 3 - Line 109: Rewrite the following sentence, the grammar is not right: “Subsequently, various content of the Al-10%Co modifier were added to prepare five sample groups with the Co content of 0%, 0.1%, 0.3%, 0.5% and 0.8%, respectively”.

Response 9: The corresponding sentence has been verified to “Definite amounts of the Al-10%Co modifier were added to form five groups of samples containing 0%, 0.1%, 0.3%, 0.5% and 0.8%, respectively.” (see Page 3-Line 110)

Point 10: Page 3 – Line 139: The units of the formula do not fit. Please, adjust the units (and formula if required) in order that the different variables fit together.

Response 10: The unit of thermal diffusivity (mm/s2) and specific heat capacity (J/(kg·K)) should be adjusted to cm2/s and J/(g·K). (see Page 4-Line 140). Moreover, the related discussions and figures have been revised.

       “83.7 mm/s2” has been verified to “0.837 cm2/s”. (see Page 9-Line 227)

       “85.6 mm/s2” has been verified to “0.856 cm2/s”. (see Page 9-Line 228)

“92.5 mm/s2” has been verified to “0.925 cm2/s”. (see Page 9-Line 231)

       The unit and values of Figure 7a have been verified. (see Page 9-Figure 7)

Point 11: Figure 2. The figure a is missing in the article! (at least in the version I have )

Response 11: The authors apologized for our carelessness. The Figure 2. (a) was added into the revised manuscript. (see Page 5-Figure 2)

Point 12: Page 7 – Line 181: .b ¿?

Response 12: The “b” has been deleted in the revised manuscript. (see Page 7- Line 182)

Point 13: Page 10 – Line 249: Rewrite the following sentence, it is confusing and not well written: “In our study, after homogenization, the thermal conductivity significantly increases due to the morphological evolution and eliminate of lattice defects”.

Response 13: The sentence has been verified to “In our study, compared with the as-cast samples, the thermal conductivities of alloys significantly increase after annealing treatment due to the evolution of morphology and elimination of lattice defects.” (see Page 10- Line 249)

Point 14: Page 13 – Line 308: “With increasing the Co content, the mechanical properties increase obviously at first, and then decrease”. Why it is so obvious? I do not see how it is so obvious At least include a reference that report that change of mechanical properties with Co content in Al-2Fe alloys.

Response 14: The authors feel very sorry for the neglectful expression. The obvious improvement of mechanical properties is owing to the grain refinement and second phase strengthening. The related references have been cited in the revised manuscript.

“When the Co content is 0.5%, the comprehensive mechanical properties reach the optimum, with the UTS, EI and HB of 138 MPa, 22.0 % and 40.5 HBW, respectively. The improvement of mechanical properties is owing to the grain refinement and second phase strengthening [3, 16].” (see Page 12 -Line 311)

Point 15: Page 14 – Line 336: “The correlation between conductivity and mechanical property”, should be “….mechanical properties”

Response 15: The subtitle has been verified to “The correlation between conductivity and mechanical properties”. (see Page 12- Line 340)

Reviewer 2 Report

It is good atricle about research of microstructures and mechanical properties of Al-2Fe-2 xCo ternary alloys with high thermal conductivity. It is found that the mechanical properties and conductivity 362 performance of the Al-2Fe alloy are simultaneously improved by Co modification. This finding is 363 favorable to develop the structural aluminum alloys with the demands of low processing cost and 364 sufficient thermal conductivity.

I have some comments.

1. I think what Section 1. Intoduction should have more detailled to show change of proterties of Al alloys after different influences. For example, authors can us 2 good papers: Konovalov, S.V., Danilov, V.I., Zuev, L.B., Filip'ev, R.A., Gromov, V.E. On the influence of the electrical potential on the creep rate of aluminum (2007) Physics of the Solid State, 49 (8), pp. 1457-1459. DOI: 10.1134/S1063783407080094; Ivanov, Y.F., Alsaraeva, K.V., Gromov, V.E., Popova, N.A., Konovalov, S.V. Fatigue life of silumin treated with a high-intensity pulsed electron beam (2015) Journal of Surface Investigation, 9 (5), pp. 1056-1059. DOI: 10.1134/S1027451015050328.
2. Figure 9. With increasing the Co content, the mechanical properties increase obviously at first, and then decrease. Need make more analysis why when the Co content is 0.5 %, the comprehensive mechanical properties reach the optimum. Why 0.5 %, but not 0.3 or 0.7?

3. Figure 7. Why situation different  if we see figure 9?

Author Response

Response to Reviewer 2 Comments

Point 1: I think what Section 1. Intoduction should have more detailled to show change of proterties of Al alloys after different influences. For example, authors can us 2 good papers: Konovalov, S.V., Danilov, V.I., Zuev, L.B., Filip'ev, R.A., Gromov, V.E. On the influence of the electrical potential on the creep rate of aluminum (2007) Physics of the Solid State, 49 (8), pp. 1457-1459. DOI: 10.1134/S1063783407080094; Ivanov, Y.F., Alsaraeva, K.V., Gromov, V.E., Popova, N.A., Konovalov, S.V. Fatigue life of silumin treated with a high-intensity pulsed electron beam (2015) Journal of Surface Investigation, 9 (5), pp. 1056-1059. DOI: 10.1134/S1027451015050328.

Response 1: Thank you for these insightful comments. These good articles provided by reviewer are very helpful for Introduction. The first paper has systematically studied the influence of electrical potential on the low-temperature creep rate of pure aluminum. The second one revealed the formation of the structure of silumin irradiated with a high-intensity electron beam, with significantly increased the fatigue life of this material. The above two articles are beneficial to formulate the preparation process. Thus, two articles are cited in Introduction. The related sentence has been added into the revised manuscript.

“The properties of aluminum alloys are controlled by some factors, such as alloying element [2, 7], preparation process [8-10], heat treatment [10, 11], etc.” (see Page 1-Line 37)

[8]. Konovalov, S. V.; Danilov, V. I.; Zuev, L. B.; Filip’Ev, R. A.; Gromov, V. E., On the influence of the electrical potential on the creep rate of aluminum. Phys Solid State 2007, 49, (8), 1457-1459.

[9]. Ivanov, Y. F.; Alsaraeva, K. V.; Gromov, V. E.; Popova, N. A.; Konovalov, S. V., Fatigue life of silumin treated with a high-intensity pulsed electron beam. J Surf Invest-X-Ray 2015, 9, (5), 1056-1059.

Point 2: Figure 9. With increasing the Co content, the mechanical properties increase obviously at first, and then decrease. Need make more analysis why when the Co content is 0.5 %, the comprehensive mechanical properties reach the optimum. Why 0.5 %, but not 0.3 or 0.7?

Response 2: After considering the comments from the reviewer, we regarded that the improvement of mechanical properties is owing to the grain refinement and second phase strengthening. According to the microstructure given in Fig. 3, the optimal microstructure of as-cast Al-2Fe-0.3Co is obtained. However, the comprehensive mechanical properties reach the optimum, when the Co content is 0.5 %. This appearance is attributed to the second phase strengthening. The as-cast Al-2Fe-0.5Co alloy contains short rod-like Al2FeCo phase, which acts as the new strengthening phase and furtherly improves the mechanical properties. When the Co content reaches to 0.8 %, the grain coarsening and plate-like Al2FeCo phases deteriorate the mechanical properties. The formation of Al2FeCo phases significantly weakens the extent of constitutional supercooling, which coarsens the α-Al grains. Moreover, the sharp of plate-like Al2FeCo phases are easy to become the origin of cracks, which decreases the mechanical properties. The related analysis was added into the revised manuscript.

“When the Co content is 0.5%, the comprehensive mechanical properties reach the optimum, with the UTS, EI and HB of 138 MPa, 22.0 % and 40.5 HBW, respectively. The improvement of mechanical properties is owing to the grain refinement and second phase strengthening [3, 16].” (see Page 12- Line 310)

[3]. Shi, Z. M.; Gao, K.; Shi, Y. T.; Wang, Y., Microstructure and mechanical properties of rare-earth-modified Al−1Fe binary alloys. Mat Sci Eng A-Struct 2015, 632, 62-71.

[16]. Luo, S.; Shi, Z.; Li, N.; Lin, Y.; Liang, Y.; Zeng, Y., Crystallization inhibition and microstructure refinement of Al-5Fe alloys by addition of rare earth elements. J Alloy Compd 2019, 789, 90-99.

       “Conversely, when the Co content reaches to 0.8 %, the grain coarsening and plate-like Al2FeCo phases deteriorate the mechanical properties. The formation of Al2FeCo phases significantly weakens the extent of constitutional supercooling, which coarsens the α-Al grains. Moreover, the sharp of plate-like Al2FeCo phases are easy to become the origin of cracks, which decreases the mechanical properties.” (see Page 12- Line 334)

Point 3: Figure 7. Why situation different if we see figure 9?

Response 3: Thank you for this valuable comment. Fig.7 mainly depicts the thermal conductivity of Al-2Fe-xCo ternary alloys. The thermal conductivities of these alloys first increase and then gradually decrease with increasing Co content. The highest thermal conductivity achieves to 208 W/(m·K) for as-cast Al-2Fe-0.3Co alloy. It is worth mention that the variations of annealed and rolled samples are similar to that of as-cast ones. Combined with the microstructure shown in Fig.3, the optimal as-cast microstructure is obtained when the Co content is 0.3%. The long needle-like primary Al3Fe phases are transformed to fine particles by Co modification, which is beneficial to the electronic transmission for free electrons. However, when the Co content is higher than 0.5%, the formation of Al2FeCo increases the volume fraction of second phase, which increases the scattering probability of free electrons and deteriorates the thermal conductivity of alloys.

Fig.9 shows the corresponding mechanical properties of the Al-2Fe-xCo alloys in various states. With the increase of Co content, the mechanical properties increase at first, and then decrease. When the Co content is 0.5%, the comprehensive mechanical properties reach the optimum. Although the optimal microstructure appears in Al-2Fe-0.3Co alloy, the short rod-like Al2FeCo phase in Al-2Fe-0.5Co alloy can act as strengthening phase, which furtherly improve the mechanical properties. However, while the Co content reaches to 0.8%, the plate-like Al2FeCo phase is easy to become the origin of cracks, which decreases the mechanical properties. The above discussion is the reason of difference between Fig. 7 and Fig. 9.

Round 2

Reviewer 1 Report

The authors have improved the article significantly. However, a general revision of the English was not performed and it is still necessary.

For instance: "alloy ingots were molten by commercial pure aluminum ingot" is not a correct English sentence. There are numerous sentences like this one and those sentences should be corrected.

Author Response

Response to Reviewer 1 Comments

Point 1: Page 1 - Line 32: “Since now, multiple… ” Since now is not the correct way to start this sentence.

Response 1: “Since now” has been revised to “For the past decade” (see Page1-Line 32 in the revised manuscript)

Point 2: Page 1 - Line 39: Rewrite the following sentence, the grammar is not right: “Specially, the alloying element with low solid solubility and forming a hard second phase in Al matrix is the ideal candidate to prepare high thermal conductivity alloys”.

Response 2: This sentence has been verified to “Specially, the alloying elements having low solid solubility in Al matrix and forming hard intermetallic phases are beneficial for enhancing the thermal conductivity and mechanical properties of alloys.” (see Page1-Line 40 in the revised manuscript)

Point 3: Page 1 - Line 41 and Page 2 – Line 42: Two consecutive sentences starting with “On the other hand,….” it is not acceptable in English.

Response 3: This sentence does be not acceptable in English. It has been verified to “Among them, this requirement can be satisfied by Fe element, on the one hand, its solid solubility in α-Al is ignorable [12], on the other hand the hard and brittle Fe-rich phases precipitate during the solidification process [13].” (see Page1-Line 42 in the revised manuscript)

Point 4: Page 2 – Line 73: Rewrite the following sentence, the grammar is not right: “However, the decrease thermal conductivity caused by Mn was quite larger than other elements because of its special configuration of extra-nuclear electron.”

Response 4: The sentence has been revised to “However, Mn element has a worse effect on the thermal conductivity than other elements because of its special configuration of extra-nuclear electron [5].” (see Page 2-Line 86 in the revised manuscript)

Point 5: Which is the annealing treatment? 24 hours at 773 K? If so, it should be clearly stated that this is the annealing treatment that the results refer to.

Response 5: Thank you for pointing out the unclear statement. In this study, the temperature and hold time of annealing treatment is 773 K and 24 h, respectively. Thus, the corresponding sentences have been verified to “The as-cast samples were longitudinally cut into three parts at 40 mm and 80 mm away from the right. The middle and right ones were homogenization treated at 773K for 24 h, and the annealed right one was rolled from a thickness of 15 mm to 3 mm with 12 passes at ambient temperature.” (see Page 3-Line 140)

Point 6: When were the samples taken after the annealing treatment and how? In this section it is only mentioned when they were taken, the methodology for the initial samples and the samples after rolling, but it is not said where and when the annealed samples presented in Figure 5 were taken.

Response 6: Thank you for this valuable comment. The annealed samples were taken in the middle ingot. After the middle ingot treated by homogenization, the corresponding testing samples were machined in the annealed ingot. To clearly state the annealing treatment, the related sentences have been verified to “The as-cast samples were longitudinally cut into three parts at 40 mm and 80 mm away from the right. The middle and right ones were homogenization treated at 773K for 24 h, and the annealed right one was rolled from a thickness of 15 mm to 3 mm with 12 passes at ambient temperature.” (see Page 3-Line 140)

Point 7: It is stated that 5 different Co levels (0%, 0.1%, 0.3%, 0.5% and 0.8%) have been produced and studied, but no results are presented for the 0.1% Co level. Why? Either remove this level from the “Materials and Methods” section, or present the results for this composition level.

Response 7: Thank you for the good advice. The results of Al-2Fe-0.1Co and corresponding sentences and figures were presented in the revised manuscript:

       “Fig. 3b-d” has been revised to “Fig. 3b-e” (see Page 6-Line 215)

       “Fig. 5d” has been revised to “Fig. 5e” (see Page 7-Line 253)

       “Fig. 5b” has been revised to “Fig. 5b and c” (see Page 12-Line 394)

“Fig. 3c” has been revised to “Fig. 3d” (see Page 12-Line 399)

The optical microstructures of as-cast, annealed and rolled Al-2Fe-0.1Co have been added into Figure 3b, 5b and 6b, respectively.

Point 8: Page 3 - Line 106: Rewrite the following sentence, it is not clear: “With the weight of about 300 g aluminium ingots were melted at 1023 K in the graphite clay crucible by electric resistance furnace”.

Response 8: This sentence has been verified to “The commercial pure aluminum ingots were melted in the graphite clay crucible by electric resistance furnace at 1023K.” (see Page 3-Line 133)

Point 9: Page 3 - Line 109: Rewrite the following sentence, the grammar is not right: “Subsequently, various content of the Al-10%Co modifier were added to prepare five sample groups with the Co content of 0%, 0.1%, 0.3%, 0.5% and 0.8%, respectively”.

Response 9: The corresponding sentence has been verified to “Definite amounts of the Al-10%Co modifier were added to form five groups of samples containing 0%, 0.1%, 0.3%, 0.5% and 0.8%, respectively.” (see Page 3-Line 136)

Point 10: Page 3 – Line 139: The units of the formula do not fit. Please, adjust the units (and formula if required) in order that the different variables fit together.

Response 10: The unit of thermal diffusivity (mm/s2) and specific heat capacity (J/(kg·K)) should be adjusted to cm2/s and J/(g·K). (see Page 4-Line 178). Moreover, the related discussions and figures have been revised.

       “83.7 mm/s2” has been verified to “0.837 cm2/s”. (see Page 9-Line 285)

       “85.6 mm/s2” has been verified to “0.856 cm2/s”. (see Page 9-Line 286)

“92.5 mm/s2” has been verified to “0.925 cm2/s”. (see Page 9-Line 289)

       The unit and values of Figure 7a have been verified. (see Page 9-Figure 7)

Point 11: Figure 2. The figure a is missing in the article! (at least in the version I have )

Response 11: The authors apologized for our carelessness. The Figure 2. (a) was added into the revised manuscript. (see Page 5-Figure 2)

Point 12: Page 7 – Line 181: .b ¿?

Response 12: The “b” has been deleted in the revised manuscript. (see Page 7- Line 228)

Point 13: Page 10 – Line 249: Rewrite the following sentence, it is confusing and not well written: “In our study, after homogenization, the thermal conductivity significantly increases due to the morphological evolution and eliminate of lattice defects”.

Response 13: The sentence has been verified to “In our study, compared with the as-cast samples, the thermal conductivities of alloys significantly increase after annealing treatment due to the evolution of morphology and elimination of lattice defects.” (see Page 10- Line 312)

Point 14: Page 13 – Line 308: “With increasing the Co content, the mechanical properties increase obviously at first, and then decrease”. Why it is so obvious? I do not see how it is so obvious At least include a reference that report that change of mechanical properties with Co content in Al-2Fe alloys.

Response 14: The authors feel very sorry for the neglectful expression. The obvious improvement of mechanical properties is owing to the grain refinement and second phase strengthening. The related references have been cited in the revised manuscript.

“When the Co content is 0.5%, the comprehensive mechanical properties reach the optimum, with the UTS, EI and HB of 138 MPa, 22.0 % and 40.5 HBW, respectively. The improvement of mechanical properties is owing to the grain refinement and second phase strengthening [3, 16].” (see Page 12 -Line 377)

Point 15: Page 14 – Line 336: “The correlation between conductivity and mechanical property”, should be “….mechanical properties”

Response 15: The subtitle has been verified to “The correlation between conductivity and mechanical properties”. (see Page 12- Line 419)
